# The Effects of a Mindfulness-Based Family Psychoeducation Intervention for the Caregivers of Young Adults with First-Episode Psychosis: A Randomized Controlled Trial

**DOI:** 10.3390/ijerph20021018

**Published:** 2023-01-05

**Authors:** Zoe Jiwen Zhang, Herman Hay Ming Lo, Siu Man Ng, Winnie W. S. Mak, Samuel Yeung Shan Wong, Karen S. Y. Hung, Cola Siu Lin Lo, Jessica Oi Yin Wong, Simon S. Y. Lui, Edmund Lin, Clara Man Wah Siu, Eric Wai Ching Yan, Sunny Ho Wan Chan, Annie Yip, Man Fai Poon, Gloria Oi Chi Wong, Jonathan Wai Hung Mak, Hillman Shiu Wah Tam, Iris Huen Hung Tse, Bobby Fook Hin Leung

**Affiliations:** 1Department of Applied Social Sciences, Hong Kong Polytechnic University, Hong Kong; 2The University of Hong Kong, Hong Kong; 3Chinese University of Hong Kong, Hong Kong; 4Castle Peak Hospital, Hospital Authority, Hong Kong; 5Kowloon Hospital, Hospital Authority, Hong Kong; 6University of West England, Bristol BS16 1QY, UK; 7School of Nursing, Hong Kong Polytechnic University, Hong Kong; 8Baptist Oi Kwan Social Sciences, Hong Kong; 9Lingnan University, Hong Kong; 10Hong Kong Family Welfare Society, Hong Kong; 11Heartfelt Listening Counselling Space, Hong Kong

**Keywords:** mindfulness-based intervention, family psychoeducation, caregivers, first-episode psychosis, randomized controlled trial

## Abstract

Objective: In this study, we investigated the effects of a mindfulness-based family psychoeducation (MBFPE) program on the mental-health outcomes of both caregivers and young adults with first-episode psychosis with an onset in the past three years through a multi-site randomized controlled trial. We also studied the outcomes of three potential mediating effects of interpersonal mindfulness, expressed emotions, and non-attachment on the program. Method: We randomly assigned 65 caregivers of young adults with psychosis to MBFPE (*n* = 33) or an ordinary family psychoeducation (FPE) program (*n* = 32); among them, 18 young adults in recovery also participated in the evaluation of outcomes. Results: Intent-to-treat analyses were conducted. No significant time × group interaction effects of MBFPE and FPE programs were found in any of the caregivers’ outcomes. However, the young adults with psychosis reported higher levels of recovery after the MBFPE program than after the ordinary FPE program (*F* = 8.268, *p* = 0.012, *d* = 1.484). They also reported a larger reduction in over-involvement of their caregivers (*F* = 4.846, *p* = 0.044, *d* = 1.136), showing that MBFPE had a superior effect to FPE in promoting recovery and reducing over-involvement. Conclusions: A brief psychoeducation program may not reduce the burden on or improve the mental-health outcome of caregivers of individuals with recent-onset psychosis. However, integrating mindfulness into a conventional family psychoeducation program may reduce the expressed emotions of caregivers, especially over-involvement. Further studies should explore how psychoeducation programs can reduce the impact of psychosis on family through sustainable effects in terms of reducing their burden and expressed emotions, using a rigorous study and adequate sample size.

## 1. Introduction

Psychosis has significant detrimental effects on the physical and social functioning of individuals [1]. Compared to the lifetime prevalence of 0.7% to 2.5% in the general population, the prevalence is much higher between the ages of 15 and 17, and the majority developed psychosis between the ages of 20 and 30 [2,3,4]. According to a review of early interventions for psychosis, many people experience serious challenges in social integration after the onset of psychosis, and the five-year relapse rate of individuals with schizophrenia could be as high as 80% [5]. Full remission is quite challenging for such individuals and around 10% of them commit suicide [5].

While young adults experience the symptoms of psychosis, they also face tremendous stigma. Accordingly, the caregivers of such young adults are likely to experience emotional burden, depression, and anxiety. The caregivers of young adults after the first onset of psychosis often struggle with performing the roles of caregiver and parent of the affected young people and maintaining the harmony of the family unit [6]. Research has summarized the main causes of caregivers’ burdens, including dealing with bizarre behaviors, negative mental-health status, and social isolation of the affected young adults [7]. Many caregivers also experience negative feelings when taking care of these young adults. For instance, they might experience grief in the face of the onset of psychosis in their family members, feelings of losing control, and even helplessness, all of which are reported to be common among caregivers [8]. They may also perceive negative responses from community members and other relatives [9], along with self-blame for the onset of psychosis in the young adults in their care because they assume it to be related to genetic problems or parental weakness to some extent [6].

The expressed emotions (EEs) of caregivers and the relationships between EEs and the prognosis of psychosis have been documented [10,11]. EEs refer to the emotional characteristics expressed by caregivers toward their family members, in terms of being hostile, critical, and over-involved in their relationships with the family members in recovery [11]. Studies have consistently shown EEs to be a significant predictor of relapse of psychosis [12]. A recent study reported that participants with schizophrenia who scored above the optimal cutoff point for criticism, hostility, and emotional over-involvement showed a 6.3-times higher 12-month schizophrenic relapse rate than those who scored below the cutoff [13]. However, other studies have reported mixed relationships between EEs and outcomes of psychosis. A 20-year prospective study found that positive symptoms increased when a high level of criticism was reported. However, EEs were not significantly associated with negative symptoms of psychosis [14]. A study among Chinese caregivers of individuals with psychosis also suggested that the emotional over-involvement (EOI) of caregivers could have negative impacts on the quality of life (QoL) of individuals with psychosis, but a similar association was not found between criticism and QoL [15].

When exploring the manifestations of EEs, it should be noted that the context and cultural norms should also be considered. Young adults lacking self-care abilities to some extent and over-involvement from their caregivers are quite common. In Asian countries, such as China, Japan, and India, strong family ties and connectedness might mean that the caregivers’ concern is likely to be converted into over-involvement [16]. A review indicated that the adjusted cutoff scores for EEs and interpretation of the EE constructs indicated that the experience of EEs could vary according to cultural norms, leading to the conclusion that there is no universal normative experience of EEs [11]. However, EE has not been explored in most family intervention studies conducted in Chinese populations [17,18] and it may be helpful to include a culturally validated measure to test whether high EE can be used to identify families who might benefit from a family intervention [19].

### 1.1. Family Psychoeducation

Family psychoeducation (FPE) is an integral part of interventions for individuals with psychosis and their caregivers [20,21]. For FPE, a cognitive behavioral approach is usually adopted to improve family functioning and teach practical skills to family members to enable them to face challenges during the initial presentations of psychosis [22]. This approach also involves skills for improving the family’s QoL in terms of multiple aspects, such as empathic understanding, resource information, and social support [9]. According to an earlier meta-analysis, the 1-year relapse rate for the FPE treatment group ranged from 6 to 12% while that for the control group was 41 to 53% [23].

However, more recent reviews reported mixed results regarding the effectiveness of FPE. A systematic review revealed that the caregivers of individuals with serious mental illnesses reported improvements in the experience of caregiving after psychoeducation programs, but the quality of evidence was very low and limited by small and heterogeneous samples [24]. A recent trial conducted in Japan also reported that FPE did not show significant results among the caregivers of individuals with a recent onset of psychosis [25]. There is still room for improvements in the efficacy and the change mechanisms of FPE interventions. Moreover, there may be a need for cultural adaptation, and the natural tendency for caregivers in collective cultures to have a high level of over-involvement should be addressed in the development and implementation of FPE programs [26].

In the past four decades, mindfulness-based programs (MBPs) have been increasingly applied for improving the well-being of individuals with chronic medical conditions. MBPs have been identified as an approach of paying attention to the present moment with a non-judgmental attitude, and in MBPs, participants learn to improve their ability to cope with stress by practicing different mindfulness exercises, including body scanning, mindful stretching, and mindful sitting [27]. During this process, the participants explore their experiences with the instructors and develop awareness and insights, which can be beneficial in terms of improved attention, regulation of emotions, and changes in cognition for stress reduction.

MBPs have been applied to support parents and caregivers in strengthening the functioning of family systems. In a study, parents and their children with mixed psychiatric diagnoses reported benefits in the mental-health outcomes of both the children and their parents, and positive changes were also found in parenting stress and parental behaviors [28]. Some studies of MBPs have been based on parents or caregivers of persons with mixed medical conditions. For example, 141 caregivers of persons with chronic conditions were randomized into an MBP or self-help control group. The participants who completed an MBP reported more reductions in depressive and anxiety symptoms and improvements in self-efficacy and mindfulness than those in the control group [29]. Although these studies provide evidence that mindfulness supports caregivers and families, there have been limitations, such as the outcomes of the care recipients not being included in the study design and high heterogeneity among participants.

Based on the above concerns, we developed a brief mindfulness-based family psychoeducation (MBFPE) program and aimed to investigate the effects of MBFPE on caregivers and the young adults in recovery (YAIR), following their first episode of psychosis. MBFPE was offered to family caregivers only, but we also invited YAIR whose caregivers participated in the MBFPE and FPE to join the study. We assessed their outcomes after the program and at a 9-month follow-up. We also planned to study the mediating roles of several potential factors in the relationships between MBFPE and the outcomes of the caregivers and YAIR. In addition to EEs, we further identified other mediators for this study. As a study of family-based mindfulness intervention reported positive changes in interpersonal mindfulness in parenting [30], it was selected as a mediator in this study. Given the association of non-attachment with various mental-health indicators, we also investigated the role of non-attachment in the intervention effects [31,32,33].

### 1.2. Objectives

This study aimed to examine the effect of MBFPE on the outcomes of both caregivers and young adults in recovery. We also investigated interpersonal mindfulness, EEs, and non-attachment as mediators in the relationships. The following hypotheses were examined:
**Hypothesis** **H1**.*Caregivers who participate in an MBFPE program will experience a reduced caregiving burden, less anxiety and depressive symptoms, less physical distress, more positive caregiving experiences, higher levels of well-being, higher levels of interpersonal mindfulness, higher levels of mindful parenting, and higher levels of non-attachment than FPE participants.*
**Hypothesis** **H2**.*YAIR whose caregivers participated in the MBFPE program will report higher levels of recovery and lower EEs than those whose caregivers participated in the FPE program.*
**Hypothesis** **H3**.*Improvements in interpersonal mindfulness, EE, and non-attachment will mediate the improvements in the caregiving burden and other outcomes in the caregivers and YAIR.*

## 2. Methods

### 2.1. Study Design

This study was designed to use mixed methods by combining quantitative and supplementary qualitative approaches to better understand the program outcomes. A two-arm randomized controlled trial was used to compare the effects between MBFPE (arm 1) and ordinary FPE (arm 2). The participants were required to complete assessments before they attended the intervention (T1) and after completion of the intervention (T2), as well as a follow-up assessment 9 months after the intervention (T3). The outcomes of the qualitative study are reported in another paper [34]. The data for the 9-month follow-up are not completed and are not included in this paper.

### 2.2. Participants

The study inclusion criteria were as follows:(1)Caregivers of young adults who had a first episode of psychosis within the last 3 years. The young adults were younger than 35 years and had forms of psychosis, including schizophrenia spectrum, bipolar, and other related psychotic disorders listed in the Diagnostic and Statistical Manual of Mental Disorders, fifth edition (DSM-5; [35]).(2)The caregivers had offered care for at least 1 year.(3)YAIR who had the capacity to provide informed consent and to respond to the questions in the assessment interviews were recruited.

The exclusion criteria of the study were as follows:(1)Caregivers who had difficulties in understanding the program contents because, for instance, they had been diagnosed with psychosis or developmental disabilities were excluded.(2)YAIR who refused to participate in regular psychiatric consultations were excluded.

Both the caregivers and YAIR participated in this project voluntarily. We recruited participants through one non-governmental organization (NGO) and two Early Assessment Service for Young People with Early Psychosis (EASY) clinics at Castle Peak Hospital and Kowloon Hospital in Hong Kong. The NGO involved in this project offers the largest number of caregiver programs in Hong Kong. Through the promotion of the research project among their members, the NGO’s social workers referred interested caregivers to the research team. The research team further promoted the project in the outpatient service units of two EASY program teams by sending research assistants to the units. With the assistance of NGO and EASY teams, we contacted and invited 174 caregivers who had applied for our caregiver program. Some caregivers were excluded from the project due to ineligibility (*n* = 26), time clashes (*n* = 46), loss of contact (*n* = 17), and lack of interest (*n* = 20). The 65 remaining caregivers were randomized into the MBFPE and FPE programs. A participant flowchart in attached in Figure 1.

We further invited the YAIR under the care of the 65 caregivers to participate in this study. Some young adults were excluded from the study because their caregivers did not want the young adults to know they had participated in the course or because they were not interested (*n* = 48). Only 18 YAIR were included in the studied sample and informed consent was obtained from all of them.

### 2.3. Procedures

After screening out the ineligible caregiver applicants, we used a computer program to randomly assign the participants to arm 1 (MBFPE) or arm 2 (ordinary FPE). All caregivers were blinded to the allocation. To reduce the potential expectancy effect, the participants were told that they would be involved in a “family psychoeducation program” without mentioning the term “mindfulness,” even if they were randomly assigned to MBFPE.

The themes and content of arms 1 and 2 were reported as a study protocol [36]. Both MBFPE and FPE were conducted face to face. The programs in both arms consisted of six sessions and the total contact time was 12 h. Arm 1 included both mindfulness training and psychoeducation for caregivers, while arm 2 included psychoeducation only. In both arms, psychoeducation was provided using a standardized video format. The content dealt with understanding psychosis, medication, treatment management, mental-health service collaboration, attention to caregivers’ experiences and distress, strategies for improving communication and problem-solving, and crisis planning, based on the best practices for working with psychosis [26,37]. The videos were contributed by multi-disciplinary mental-health professionals, namely a psychiatrist, a clinical psychologist, two psychiatric nurses, an occupational therapist, three social workers in integrated mental-health services, three caregivers, and four YAIRs. The videos were supplemented by discussion and sharing by participants in MBFPE and FPE.

In arm 1, the first hour was spent on mindfulness training. Qualified mindfulness-based instructors introduced mindfulness exercises, including body scanning, mindful stretching, mindful walking, mindful sitting, mindfulness with difficult moments, and befriending, according to the sequences in a typical eight-week mindfulness program [38,39]. Five 10 min audio files were sent to participants as homework after each session. The second hour of each session was spent on psychoeducation. The instructors viewed the videos and facilitated discussion and sharing among participants. In arm 2, psychoeducation instructors used the whole session to show video, answer questions, and facilitate sharing among the participants. The session outline for both arms is included in Appendix A.

The caregivers who completed the data collection were given HKD 100 (around USD 12) cash coupons at T2 and T3 and the YAIR were given HKD 100 (around USD 12) coupons when they completed the assessments at T1, T2, and T3. A cash remuneration coupon is a commonly used incentive in research to promote engagement [40].

The caregivers were told that the study was independent from the healthcare service and they were able to withdraw from the study at any time without any negative consequences. This project was registered with the United States Clinical Trials Registry (NCT03688009).

### 2.4. Measures

**Caregivers’ burden**. The Zarit Burden Interview (ZBI; [41]) was used to measure the caregivers’ burden in this study. It contains 22 items that focus on the perceived stress of caregivers. The burden level of caregivers is measured across five aspects, namely their health status, social life, financial status, psychological well-being status, and relationship with the family member in recovery. The caregivers state their levels of discomfort in response to the items by choosing the extent on a scale from 0 (“not at all”) to 4 (“extremely”). A higher score indicates a potential higher level of caregiver burden. The Cronbach’s alpha in this study was 0.933.

**Caregiving experiences**. The Experience of Caregiving Inventory (ECI; [42]) was adopted in this study to test caregiving experiences. According to the research purpose, we selected three subscales, namely the subscale of stigma, the subscale of effects on the family, which was used to evaluate the negative impact of illness on family life, and the subscale of positive experience in caregiving. These three subscales contain 26 items in total. In this study, the Cronbach’s alpha was 0.793.

**Caregivers’ physical distress**. We used the subscale of physical distress in the Body–Mind–Spirit Well-Being Inventory (BMSWBI; [43]) to measure the physical health of the caregivers. The subscale measures physical symptoms in the last week, such as fatigue and headache. It is a self-reported scale, with 14 items rated on a scale ranging from 0 (“no distress at all”) to 10 (“extreme distress”). The Cronbach’s alpha was 0.948, showing a high internal consistency in this study.

**Mental-health status of caregivers**. The mental-health status of the caregivers was measured using the Hospital Anxiety and Depression Scale (HADS; [44]). This self-reported scale has seven items for depression and seven items for anxiety, with the score of each item ranging from 0 (“low”) to 4 (“severe”). In this study, the Cronbach’s alphas of HADS were 0.745 for depression and 0.851 for anxiety.

**Well-being status of caregivers**. The WHO-5 Well-Being Index [45] is a well-established measurement for measuring psychological well-being. It contains five self-reported items and is used for caregivers to recall their well-being status in the past 2 weeks. The rating for each item ranges from 0 (“at no time”) to 5 (“all of the time”), with higher scores indicating a higher subjective perception of well-being status. The Cronbach’s alpha was 0.947 in this study, showing a high internal consistency.

**Interpersonal mindfulness of caregivers**. We used the Interpersonal Mindfulness in Parenting Scale (IM-P, [46]) to measure the interpersonal mindfulness of the caregivers. The scale has been validated in a Hong Kong Chinese sample. The Chinese version of IM-P used has 23 items [47] and it showed good internal consistency in this study (Cronbach’s alpha = 0.852).

**Caregivers’ non-attachment**. To assess the psychological and social adaptation of the caregivers, we adopted the Non-Attachment Scale [31]. The Chinese short form of the Non-Attachment Scale (NAS-SF) is a self-reported scale containing eight items. Each item is scored from 1 (“strongly disagree”) to 6 (“strongly agree”) [48]. A good internal consistency of 0.91 was reported in this study.

**Recovery level of young adults**. The Mental Health Recovery Measure (MHRM; [49]) was used in this study to measure the mental-health recovery status of the young adults based on their experience with psychosis. This scale measures multiple aspects of recovery status, such as overcoming stagnation, self-empowerment, and new potential. It is scored on a five-point scale ranging from 1 (“totally disagree”) to 5 (“totally agree”). A higher score indicates a higher level of recovery. The MHRM showed a good internal consistency of Cronbach’s alpha (0.944 in this study).

**Young adults’ perceptions of their caregivers’ expressed emotions**. The Level of Expressed Emotion Scale (LEES; [50]) was applied in this study. LEES is a validated 12-item scale used by young adults with psychosis to self-report their family’s EEs from the aspects of criticism, hostility, and over-involvement. Each item is scored on a 5-point scale, with a higher score indicating a higher level of EEs. The internal consistency (Cronbach’s alpha) of the whole LEES was reported to be 0.924 in this study, and the values for criticism, over-involvement, and hostility were 0.744, 0.854, and 0.938, respectively.

**Fidelity of MBFPE**. To ensure the fidelity of the study, all of the sessions were audio-recorded and 20% of the clips were randomly selected and assessed by independent raters. According to the protocol, an independent rater examined the quality and consistency of implementation of the intervention protocol. The fidelity of arm 1 was further examined using the Mindfulness-Based Interventions: Teaching Assessment Criteria Scale (MBI: TAC; [51]).

**Dosage and participant satisfaction**. In a post-group survey, participants of MBFPE and FPE were invited to rate their level of satisfaction using a 4-point scale (1 = very dissatisfied, 4 = very satisfied). The caregivers who participated in the MBFPE were further invited to report their time (in minutes per day) spent on mindfulness practice.

### 2.5. Data Analysis

After data collection through the randomized controlled trial, we used the intent-to-treat approach [52] to conduct the analysis. Multiple imputation methods were used to manage missing data [53]. We included participants who completed 50% of the MBFPE or FPE sessions or above to complete the post-test and follow-up to evaluate the program. Within-group analyses were conducted to investigate the effects of MBFPE and FPE. We used 2 × 2 analysis of variance (ANOVA) to examine the between-group effects of arm 1 and arm 2. In view of the possible baseline difference in mental-health status of participants, we controlled our primary outcome measure (caregiver burden) in subsequent data analysis of all other outcomes of the participants and YAIR. We further conduct subgroup analysis to investigate outcomes based on caregiver’s demographic variables using t-tests. A two-sided *p*-value of 0.05 or less was considered statistically significant. The effect sizes were calculated and the interpretations of within-in group changes were based on the views of Cohen [54], which suggested that *d* = 0.2, 0.5, 0.8 can be considered a small, medium, and large effect size. All of the quantitative analyses were performed using SPSS version 27.0.

## 3. Results

### 3.1. Baseline Analyses of the Participants

Among the 65 caregivers, 78.5% were female and around two-thirds (64.6%) were aged 51 years or older. As for their education level, about half (49.2%) of the caregivers had obtained secondary education and more than a third (36.9%) had completed tertiary education. The majority of the caregivers (70.8%) were married, and a similar percentage (66.2%) had religious beliefs. About a third of the caregivers (33.8%) had full-time jobs, and most of the caregivers (89.2%) were living with the YAIR. Table 1 summarizes the profiles of the caregivers and shows that there were no significant differences in demographic variables between MBFPE and FPE participants.

Only 18 YAIRs participated in the study. We analyzed the group differences between the young adults who had their caregivers participate in MBFBE and FBE in the study. Table 2 summarizes the profiles of the young adults who had their caregivers participate in MBFPE and FPE and shows that two groups had no significant difference in their gender, age, marriage status, religion, job status, or clinical status, including their diagnosis, diagnosis duration, and family history of psychiatric disorders.

In view of the small number of YAIR who participated in the study, we further conducted a baseline comparison of the caregiver profiles and investigated whether there was a significant difference between the young adults who participated in the study and those who did not. We compared their demographic profile (gender, age, education, marital status, religion, employment) and psychiatric history (time of onset and having family history of psychiatric disorders) and found significant differences in their education level (Χ^2^ = 7.194, *p* = 0.027) only. The pretreatment conditions of caregiver’s three major mental-health indicators (caregiver burden, depression, and anxiety) did not differ significantly between the YAIR participating in the study and the non-participating group (all *ps* > 0.05).

### 3.2. Within-Group Effects

#### 3.2.1. Within-Group Effects of Mindfulness-Based Family Psychoeducation

Among the caregivers in the MBFPE group, small-to-large effect sizes were observed over time, as shown in the within-group data analysis in Table 3. After controlling the covariance in caregiver burden, we analyzed the outcomes and found that anxiety (*d* = 0.201) and positive caring experience (*d* = 0.286) had small effect sizes. Reductions in physical distress (*d* = 0.659) and depression (*d* = 0.565) had medium effect sizes, and the effect on family, the subscale of caregiving experiences (*d* = 0.804, *p* = 0.032), showed a significant large-sized improvement. The caregivers reported changes in all outcomes in the expected directions, except caregiver burden and non-attachment, which showed very mild deterioration after MBFPE.

Regarding the YAIR in the MBFPE group, reductions were observed in the LEES total score and all three subscale scores. The reductions in over-involvement (*d* = 0.644) and hostility (*d* = 0.670) reflected medium sizes of improvements in EE. YAIR further reported a large size of improvement in recovery (*d* = 1.391), but the changes did not reach the level of statistical significance (see Table 4 for details).

#### 3.2.2. Within-Group Effects of Ordinary Family Psychoeducation

The caregivers’ burden (*d* = 0.238), effects on the family (*d* = 0.429), well-being (*d* = 0.238), physical distress (*d* = 0.327), depression (*d* = 0.220), and non-attachment (*d* = 0.211) showed positive improvements with small effect sizes. No significant within-group effects were found among the examined outcomes of the caregivers in the FPE group. All outcomes of the caregivers over time are summarized in Table 3.

No significant changes were found among YAIR in the FPE group. The YAIR reported an improvement in recovery with a small effect size (*d* = 0.352). Unexpectedly, the YAIR reported non-significant increases in LEES total score, the over-involvement and hostility subscale scores. For details of the outcomes, please refer to Table 4.

### 3.3. Between-Group Effects

We used 2 × 2 ANOVA to investigate the time × group interaction effects of the MBFPE and FPE programs. As shown in Table 3, no statistically significant interactions were found in the data related to caregivers. After controlling the covariance in caregiver burden, MBFPE showed a small but non-significant superior effect on physical distress (*d* = 0.278).

The outcomes in YAIR showed time × group interactions with large effect sizes on the LEES over-involvement (*F* = 4.846, *p* = 0.044, *d* = 1.136) and on recovery (*F* = 8.268, *p* = 0.012, *d* = 1.484). The details are reported in Table 4.

### 3.4. Subgroup Analysis

We further investigated the individual differences in changes in the caregivers and YAIR and attempted to identify significant predictors of program outcomes by combining the participants in the two arms. Using paired-sample t-tests, we found that male caregivers had a marginally larger improvement in well-being than female caregivers (*t*–1.928, *df* = 64, *p* = 0.058, *d* = 0.606). We also found that caregivers with a secondary or lower level of education had a significantly larger improvement in well-being than caregivers with a tertiary level of education (*t* = 2.344, *df* = 64, *p* = 0.022, *d* = 0.598).

### 3.5. Dosage, Participant Satisfaction, and Program Fidelity

Among the participants who received the MBFPE, half (*n* = 15, response rate 50.0%) responded to the question about time spent on mindfulness exercises per day. The average time spent was 10.3 min (SD = 5.5 min).

All of the caregivers (*n* = 65) responded to our satisfaction survey. Based on a 4-point scale, caregivers from the MBFPE group gave a mean satisfaction score of 3.29 (SD = 0.41), and those from the FPE group gave a score of 3.46 (SD = 0.34).

An independent assessor who had completed professional training in mindfulness-based cognitive therapy and had 10 years of experience with teaching mindfulness-based interventions conducted a fidelity test using MBI: TAC. The average rating for the MBFPE program, on a 6-point scale, was 3.75 (SD = 0.81, range 3.0–5.0).

## 4. Discussion

Studies have demonstrated the effective use of mindfulness training in reducing caregiver stress. Our research team conducted a randomized controlled trial involving Chinese caregivers, based on an 8-week benchmark mindfulness-based program and caregivers of people with mixed medical conditions [29]. In this study, we aimed to investigate whether such positive outcomes could be replicated in the caregivers of young adults in recovery following first-episode psychosis after a brief mindfulness-based program. We tested the effects of the program on both the caregivers and the young adults in recovery. We expected the findings to answer questions caused by the limitations in the literature, especially with regard to the involvement of young adults with psychosis and the use of expressed emotions as a potential mediator of outcomes.

We did not find evidence to support Hypothesis 1. No significant differences were found in all caregivers’ outcomes between the MBFPE and FPE groups. When we focused on within-group changes, there were still no significant results found in either of the arms. We noticed that the caregivers reported improvements with small to large effect sizes in effects on family, positive caring experience, physical distress, depression, and anxiety after MBFPE. Only the improvement in effect on family reached the level of statistical significance. The caregivers who received conventional FPE reported improvements with small effect sizes in the caregiving burden, effect on family, physical distress, well-being, and non-attachment, but none of them reached the level of statistical significance. The time × group interaction effects of the MBFPE and FPE groups did not show significant changes in any of the caregivers’ outcomes. We did not replicate the findings of a recent systematic review [55], which reported that mindfulness-based interventions could produce superior effects on caregiver outcome from psychoeducation programs. This suggests that the effects of mindfulness-based interventions might depend on their target population, duration, structure, components, and other factors. Our study finding is consistent with a recent trial in Japan, which found that family psychoeducation did not show positive effects in the caregivers of people with recent-onset psychosis, defined as a duration of less than 5 years [22].

There may be three reasons for the absence of superior effects of the mindfulness-based psychoeducation program in our study. First, our restrictive inclusion criteria meant that the caregivers in our study had experienced a family transition, as the onset of their family members’ psychosis was within the last 3 years. Compared with caregivers who experienced chronic caregiving stress, the caregivers in our study may have felt overwhelmed by the drastic changes in the mental condition of their family members after the onset of psychosis and the subsequent adjustments made by the entire family. For individuals who experience major losses or life changes, learning and practicing mindfulness exercises can be challenging, as strong emotions may surface during periods of silence [38]. Further studies may consider adjusting the inclusion criteria to include caregivers whose family member’s onset of psychosis occurred in the last 3 to 10 years. It is likely that these families may benefit more from a brief psychoeducation program when their family members with psychosis are in a relatively stable condition. The low response rate about time spent on mindfulness exercises in the post-group survey suggests that some participants did not practice mindfulness exercises at home. For mental-health care practitioners, it may also be helpful to clearly explain the components of the psychoeducation programs and allow caregivers to choose the program that they prefer.

Second, our program structure was based on the principle that the intervention group and active control would have identical contact hours, i.e., 12 h over six sessions. However, the instructors in the project reflected that the caregivers in the active control group were allowed to reflect on and discuss the psychoeducation videos with adequate time allocated for social and emotional support, while for the mindfulness group, the instructors may have felt restricted in spending time to address the emotional needs of the participants, as the mindfulness exercises could have taken up half of the program time. Further studies should pay attention to the development of the program content and ensure that caregivers have adequate time and understand and apply their newly learned knowledge and skills.

Third, the expressed emotions and caregiver burden experienced by family caregivers and people in recovery may drastically increase due to the suspension of mental-health care and social services during the COVID-19 pandemic and the increase in family members’ time spent at home. A survey of a Hong Kong Chinese sample reported a high expressed emotion prevalence of 63% among individuals with a diagnosis of schizophrenia, which was much higher than the previous reported prevalence range of 30–40% [13,56,57]. A significant increase in expressed emotions may strongly increase the caregiver burden and mental-health symptoms of caregivers, and a brief psychoeducation program may not be adequate to mitigate their mental distress. The COVID-19 pandemic also increased the difficulty of recruitment in our study, and many families in need were reluctant to participate in a face-to-face study. We discuss this issue further in the Limitations section.

Subgroup analysis revealed that male caregivers and caregivers with a secondary or lower level of education had larger improvements in well-being. Male caregivers have been found to have lower caregiver burden at pretest, and they may be more responsive than female caregivers to a brief psychoeducation program. The reason for the better outcomes of caregivers with a lower (vs. higher) education level is unclear. One possibility is that parents in our study who had a higher education level were more likely to invest time and resources in their children’s development and, thus, it was more difficult for them to accept their children’s mental-health challenges. Further studies should explore the relationship of caregiver’s education level with caregiver burden and expressed emotions and the implications in family intervention outcome.

The results of analysis of the young adults in recovery in this study partially supported Hypothesis 2. After MBFPE, the young adults reported improvements in over-involvement and hostility with medium effect sizes and an improvement in the level of recovery with a large effect size. Unexpectedly, the young adults reported higher levels of perceived over-involvement, hostility, and overall expressed emotions after their caregivers completed an ordinary FPE, although their recovery score also slightly increased. The time × group interaction effects of the MBFPE and FPE programs were significant in terms of the differences in recovery levels of the young adults and the perceived over-involvement, showing that MBFPE had a superior effect to FPE in promoting recovery and reducing over-involvement. Such findings are consistent with our qualitative data analysis, as many caregivers shared that they learned not to interfere with the young adults in recovery, especially when they felt worried and guilty about the family member’s condition [31]. This shows that caregivers may feel pressure when they learn about psychosis and want to do something to improve the condition of the young adults with psychosis after psychoeducation. However, mindfulness may help caregivers to manage their expectations and accept that the recovery of psychosis is slow and out of their control. Therefore, it is important for caregivers to accept the illness of their family members and to manage their own emotions mindfully.

Although the sample size of the young adults in recovery was small, their perception of the expressed emotions and over-involvement was largely reduced, and their recovery level was found to significantly increase after their caregivers completed the MBFPE program. If we focus on the outcome of expressed emotions, MBFPE played a greater role in reducing over-involvement than in reducing criticism and hostility. A previous Chinese study reported that the over-involvement of caregivers significantly influenced the QoL of individuals with psychosis [15], which might explain this finding in our study. However, according to the views of previous studies [11,14], the levels of expressed emotions might not be stable over time. The long-term effects of MBPs in this study are uncertain and can only be determined during the data analysis of the follow-up effects of the interventions when we complete the 9-month follow-up data collection. As for Hypothesis 3, we were not able to analyze the mediating effect due to the results from the caregivers and the small sample size of the young adults.

## 5. Limitations and Implications

First, we encountered many difficulties in recruitment due to the impact of the COVID-19 pandemic. Although our research team had spent around 20 days stationed in the outpatient units of two EASY program teams, few young adults and caregivers responded to our invitations after the waves of COVID-19. Many of them explicitly expressed their reservations about participating in a face-to-face caregiver program. The young adults were even less responsive to our invitations than the caregivers, although we offered incentives for participation. It is not certain that the results of our study can be generalized to other populations, particularly young adults in recovery, given the small sample. Further studies should develop more effective strategies to recruit participants, especially in collecting consent to use clinical records as one of the outcomes of the study.

Second, in this paper we focus on reporting the immediate effects of the program outcomes. We still have not completed our follow-up data collection. It is uncertain whether the program will have sustainable effects 9 months after completion. Our research team plans to report the findings in a later manuscript. Last, the assessment of expressed emotions in this study was based on the perceptions of the young adults in recovery. We did not measure the perceptions of the caregivers. It is interesting that after the ordinary psychoeducation program, the ratings for over-involvement and hostility increased, while in the MBFPE group, these ratings decreased. Mindfulness skills might allow caregivers to gain new insights with non-judgmental attitudes into young adults with psychosis. Further studies of MBPs may apply different measures, such as the five-minute speech sample, a well-established behavioral coding to measure EEs [58], and investigate its mediating effects on the outcomes of MBPs.

## 6. Conclusions

A brief psychoeducation program may not reduce the burden and mental-health outcome of caregivers of individuals who have a resent onset of first-episode psychosis. Caregivers experience serious challenges while providing care, and mental-health professionals should ensure that they receive adequate support that meets their needs. In such a situation, a brief family intervention may not be adequate to support their needs in this critical period. The preliminary evidence indicating that a mindfulness-based intervention can modify expressed emotions, especially over-involvement, is encouraging. Further studies should explore how psychoeducation programs can reduce the impact of psychosis on the family with sustainable effects in reducing their burden and expressed emotions using a rigorous study design and adequate sample size.

## Figures and Tables

**Figure 1 ijerph-20-01018-f001:**
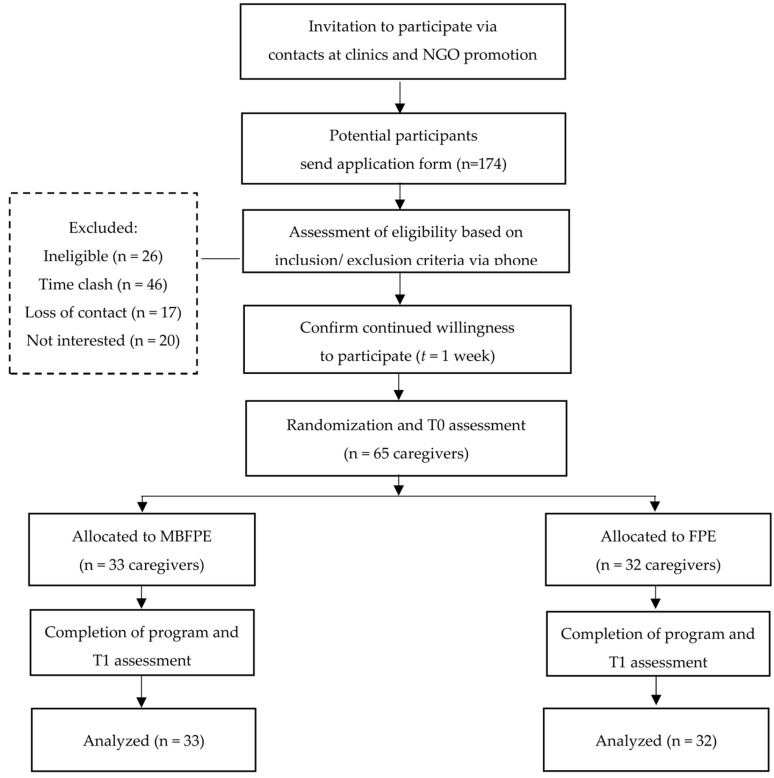
Flow diagram of the process for randomized controlled trial.

**Table 1 ijerph-20-01018-t001:** Baseline comparison of caregivers (*n* = 65).

	MBFBE (*n* = 33)		FBE (*n* = 32)		*t*	*Χ* ^2^	*p*
Variables	*n*	Percent	*n*	Percent			
**Gender** Male Female	7 26	21.2 78.8	7 25	21.9 78.1		0.004	0.948
**Age** <40 40–50 51–60 >60	4 8 17 4	12.1 24.2 51.5 12.1	4 7 19 2	12.5 21.9 59.4 6.3	−0.197		0.844
**Education** Below Primary Primary Secondary Tertiary	2 4 17 10	6.1 12.1 51.5 30.3	- 3 15 14	- 9.4 16.9 43.8		2.920	0.404
**Marriage** Single Married Separated Widowed	5 21 6 1	15.2 63.6 18.2 3.0	4 25 2 1	12.5 78.1 6.3 3.1		2.444	0.485
**Religion** No Christianity Buddhism Other	23 6 3 1	69.7 18.2 9.1 3.0	20 10 2 -	62.5 31.3 6.3 -		2.394	0.495
**Job** Unemployed Searching Retired Part-time Full-time	7 7 5 4 10	21.2 21.2 15.2 12.1 30.3	4 7 8 1 12	12.5 21.9 25.0 3.1 37.5		3.945	0.557
**Live together** Yes No	29 4	87.9 12.1	29 3	90.6 9.4		0.128	0.721
**Number of Family Member** 1–2 3–4 5–6	6 21 6	18.2 63.6 18.2	4 22 6	12.5 68.8 18.8	0.306		0.760
**Hour of Caregiving** <10 h 10–20 h >20 h	26 5 2	78.8 15.2 6.1	24 6 2	75 18.8 6.3	0.803		0.425

**Table 2 ijerph-20-01018-t002:** Baseline comparison of demographic and mental-health conditions of young adults in recovery (*n* = 18).

	MBFBE (*n* = 8)		FBE (*n* = 10)		*t*	*Χ*	*p*
Variables	*n*	Percent	*n*	Percent			
**Gender** Male Female	4 4	50.0 50.0	5 5	50.0 50.0		0.000	1.000
**Age** <20 20–30 >30	2 5 1	25.0 62.5 12.5	2 5 3	20.0 50.0 30.0	0.966		0.349
**Education** Secondary Tertiary	3 5	37.5 62.5	1 9	10.0 90.0		1.945	0.163
**Marriage** Single Married	8 -	100 -	9 1	90.0 10.0		0.847	0.357
**Religion** No Christianity	6 2	75.0 25.0	7 3	70.0 30.0		0.055	0.814
**Job** Unemployed/searching Part-time Full-time	7 1 0	87.5 12.5 0.0	7 0 3	70.0 0 30.0		3.825	0.281
**Diagnosis** Schizophrenia Psychosis	5 3	62.5 37.5	6 4	60.0 40.0		0.012	0.914
**Diagnosis Duration (month)** <12 12–24 >24	3 1 4	37.5 12.5 50.0	5 3 2	50.0 30.0 20.0	−1.701		0.108
**Family History of Psychiatric Disorders** No Yes	6 2	75.0 25.0	6 4	60.0 40.0		0.450	0.502

**Table 3 ijerph-20-01018-t003:** Measures over time for the MBFPE group and FPE group on outcomes of caregivers (caregiver burden as covariates).

	MBFBE (*n* = 33)				FBE (*n* = 32)						
Variables	Pretest	Posttest	*d*	*p*	Pretest	Posttest	*d*	*p*	*Time* *F, p, d*	*Group* *F, p, d*	*Time X Group* *F, p, d*
Caregiver burden	39.52 (13.83)	39.70 (15.31)	0.031	0.931	42.09 (16.94)	40.81 (15.01)	0.238	0.511	0.149, 0.701, 0.090	0.275, 0.602, 0.127	0.264, 0.609, 0.127
Stigma	8.52 (4.62)	7.94 (4.64)	0.043	0.905	8.97 (3.94)	8.75 (4.07)	0.028	0.938	0.026, 0.872, 0.041	0.114, 0.737, 0.090	0.223, 0.639, 0.127
Effect on Family	10.03 (5.69)	9.79 (5.83)	0.804	0.032	10.75 (5.04)	9.88 (4.63)	0.429	0.251	5.481, 0.022, 0.594	0.001, 0.978, 0.007	0.135, 0.714, 0.090
Positive caring experience	30.42 (7.04)	31.36 (8.14)	0.286	0.438	28.75 (6.32)	29.00 (7.56)	0.127	0.716	0.083, 0.774, 0.063	1.527, 0.221, 0.314	0.179, 0.673, 0.110
Physical distress	27.94 (21.97)	27.85 (19.41)	0.659	0.075	37.66 (33.80)	30.63 (25.72)	0.327	0.379	3.766, 0.057, 0.492	0.775, 0.382, 0.220	1.222, 0.273, 0.278
Depression	7.18 (3.54)	6.33 (3.71)	0.565	0.126	6.75 (4.54)	5.72 (4.50)	0.220	0.557	0.251, 0.618, 0.127	1.115, 0.295, 0.271	0.012, 0.914, 0.028
Anxiety	7.76 (3.29)	7.06 (3.48)	0.201	0.574	8.59 (3.97)	7.22 (3.78)	0.168	0.655	0.000, 0.987, 0.004	0.080, 0.778, 0.063	0.560, 0.457, 0.191
Well-being	12.94 (5.62)	14.33 (4.90)	0.014	0.970	13.56 (5.22)	14.19 (3.87)	0.238	0.524	0.265, 0.608, 0.127	0.451, 0.504, 0.168	0.354, 0.554, 0.155
Interpersonal mindfulness	77.03 (11.62)	78.45 (8.90)	0.090	0.785	77.41 (9.72)	79.22 (9.97)	0.063	0.902	0.007, 0.933, 0.021	0.422, 0.519, 0.168	0.018, 0.892, 0.034
Non-attachment	31.55 (8.56)	31.52 (8.40)	0.168	0.649	32.75 (6.93)	32.84 (6.30)	0.211	0.565	0.516, 0.475, 0.180	0.919, 0.342, 0.247	0.000, 0.997, 0.000

Note: *d*: Cohen’s d, *F*: F score for Repeated Measure ANOVA.

**Table 4 ijerph-20-01018-t004:** Measures over time for the MBFPE and FPE groups on outcomes of young adults in recovery (caregiver burden as covariates).

	MBFBE (*n* = 8)				FBE (*n* = 10)						
Variables	Pretest	Posttest	*d*	*p*	Pretest	Posttest	*d*	*p*	*Time* *F, p, d*	*Group* *F, p, d*	*Time X Group* *F, p, d*
Expressed emotions (total score)	26.38 (9.49)	25.25 (11.21)	0.063	0.930	30.60 (9.32)	34.10 (10.43)	0.063	0.926	0.008, 0.928, 0.063	3.409, 0.085, 0.953	1.627, 0.221, 0.659
Criticism	9.38 (3.74)	9.25 (4.30)	0.063	0.941	12.20 (2.30)	12.10 (3.41)	0.011	0.988	0.005, 0.945, 0.036	6.663, 0.021, 1.334	0.001, 0.982, 0.012
Over-involvement	8.88 (3.04)	8.25 (4.10)	0.644	0.460	8.60 (3.53)	11.00 (3.56)	0.063	0.918	0.220, 0.646, 0.238	0.699, 0.416, 0.434	4.846, 0.044, 1.136
Hostility	8.13 (3.60)	7.75 (3.88)	0.670	0.444	9.80 (4.47)	11.00 (4.40)	0.090	0.911	0.027, 0.871, 0.090	3.090, 0.099, 0.908	0.715, 0.411, 0.439
Recovery	108.13 (7.86)	117.50 (13.46)	1.391	0.139	112.40 (19.93)	113.00 (19.24)	0.352	0.635	3.694, 0.074, 0.994	0.224, 0.643, 0.247	8.268, 0.012, 0.1484

Note: *d*: Cohen’s d, *F*: F score for Repeated Measure ANOVA.

## Data Availability

The raw data supporting the conclusions of the article will be made available by the authors upon request.

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
