# Peer review of "The Effects of a Mindfulness-Based Family Psychoeducation Intervention for the Caregivers of Young Adults with First-Episode Psychosis: A Randomized Controlled Trial"

_ijerph, 2023, doi:10.3390/ijerph20021018_

Round 1
Reviewer 1 Report
The manuscript is overall interesting and assess the important topic of hepling families and reducing family burden through psychoeducation intervention.
I have however some concerns and some issue that should be resolved in order to improve the quality of the manuscript.
1- The methods should be slightly modified. The description of the measures is very long and comprehansive, but I believe it can be at least partially moved into the Supplementary. Similarly, the analyses part should be increased: in this sense, I do not understand how confoudners/covariates have been included (if they have) in the analyses. I suggest increasing the Analyses part and clrearly defining all aspects in order to foster replicability.
2- The tables should be clearly revised since I spotted some weird numbers/typos. For example, in Table 1 the parcentage of Live Together "Yes" is reported to be 27.9% and I believe it should be 87.9%. Similarly, in the text the table are not always clearly defined: as an example, the sentence "Among the caregivers in the MBFPE group, there were small-to-large effect sizes over time, as shown in the within-group data analysis in Table 1" I believe it refers to other table, since table 1 reports only demographic aspects.
3- I suggest reporting clearly in the abstract the population of interest. The abstract reports "young adults with psychosis" but it does not specify better. Please, specify that they are recovering from a First-Episode - this is very important! Similarly, please report if the type of treatment the young psychotic patients were undergoing during the period of study.
4- A full revision of language is needed, since few typos have been spotted throughout the text
Author Response
The manuscript is overall interesting and assess the important topic of helping families and reducing family burden through psychoeducation intervention.
I have however some concerns and some issue that should be resolved in order to improve the quality of the manuscript.
1- The methods should be slightly modified. The description of the measures is very long and comprehensive, but I believe it can be at least partially moved into the Supplementary. Similarly, the analyses part should be increased: in this sense, I do not understand how confounders/covariates have been included (if they have) in the analyses. I suggest increasing the Analyses part and clearly defining all aspects in order to foster replicability.
Our responses:
thank you for your comment. We have made the following revisions in this version: 1) We have expanded the second paragraph in procedure (line 243-250) and added a third paragraph (line 258-266) to explain the structure of the mindfulness-based family psychoeducation program. Besides, an appendix on the session themes for both mindfulness arm and psychoeducation arm are attached at the end of the manuscript for interested researchers to replicate the study.
2) the description of measures is edited.
3) in data analysis, additional information is added to explain the co-variants in data analysis and its rationale. As indicated in both table 3 and 4, caregiver burden is the only co-variants. We also expanded the data analysis section and included subgroup analysis to investigate the individual differences in caregiver outcome.
2- The tables should be clearly revised since I spotted some weird numbers/typos. For example, in Table 1 the percentage of Live Together "Yes" is reported to be 27.9% and I believe it should be 87.9%. Similarly, in the text the table are not always clearly defined: as an example, the sentence "Among the caregivers in the MBFPE group, there were small-to-large effect sizes over time, as shown in the within-group data analysis in Table 1" I believe it refers to other table, since table 1 reports only demographic aspects.
Our responses: thank you for pointing out the typos:
We have made the following revisions:
1) in table 1, % of caregivers living together with young adults in psychosis should be 87.9%
2) Among the caregivers in the MBFPE group, there were small-to-large effect sizes over time, as shown in the within-group data analysis should be Table 3 (now line 389)
3- I suggest reporting clearly in the abstract the population of interest. The abstract reports "young adults with psychosis" but it does not specify better. Please, specify that they are recovering from a First-Episode - this is very important! Similarly, please report if the type of treatment the young psychotic patients were undergoing during the period of study.
Our responses: thank you for your suggestion: In abstract, we have added such information in the study objective as follows: In this study, we investigated the effects of mindfulness-based family psychoeducation (MBFPE) on the mental health outcomes of both caregivers and the young adults with first-episode psychosis who had onset in recent three years, using a multi-site randomized controlled trial.
In this project, young people who were not attending regular psychiatric consultation were excluded from the project and we have included this in exclusion criteria (line 186-187).
4- A full revision of language is needed, since few typos have been spotted throughout the text
Our responses: thank you for your reminder. We have proofread the full manuscript and make sure the language is meeting the standard for publication. We have sent the manuscript for native speaking editor for professional service.
Thank you for your useful comments and they have largely improved the overall quality of the manuscript.
Reviewer 2 Report
GENERAL COMMENT
Thank you for allowing me to review this interesting paper. the issue of the suffering and fatigue of caregivers of psychiatric young adults is particularly relevant and requires the identification of effective strategies to limit the negative effects of emotional burden.
The present work analysed the short-term effects of a mindfulness-based invention on a group of caregivers, divided into two experimental groups. The study also attempted to share the children of these caregivers, but encountered many difficulties. The intervention, despite its solid scientific basis, did not find many significant results, but often found an improvement (not statistically significant) in the dimensions considered.
COMMENT
I believe there are some limitations in the present study.
Firstly, there is no description of the intervention as well as the duration of the intervention. How was it delivered? In presence or remotely? How long did each individual session last and how long did the intervention itself last? How many sessions had to be attended for a minimum number of participants to be reached, which was adequate to evaluate the effectiveness or otherwise of the intervention? In some studies, for example, this limit is set at 75% of the meetings (there are numerous examples in the literature).
I think this element could be very relevant to identify those participants who actually attended an adequate number of sessions and those who did not reach a minimum number to be effective.
In addition, it could also be interesting to look for significant differences between the participants in the different experimental groups with respect to the dependent variables considered.
Without knowing what type of intervention was carried out, for how long, and without assessing further differences in the sample, I do not think it is possible for me to pass judgement on the work. With regard to the introduction, on the other hand, I think it is well written, although I would suggest dividing it into paragraphs and especially giving more prominence to the part concerning cultural differences in EE. Regarding discussion and conclusion, I believe that Authors could be find more reasons why the intervention showed limited effectiveness after considering the suggested indicators.
I wish you good luck and good work.
Author Response
I believe there are some limitations in the present study.
Firstly, there is no description of the intervention as well as the duration of the intervention. How was it delivered? In presence or remotely? How long did each individual session last and how long did the intervention itself last? How many sessions had to be attended for a minimum number of participants to be reached, which was adequate to evaluate the effectiveness or otherwise of the intervention? In some studies, for example, this limit is set at 75% of the meetings (there are numerous examples in the literature).
I think this element could be very relevant to identify those participants who actually attended an adequate number of sessions and those who did not reach a minimum number to be effective.
Our responses: thank you for your comment. In view of both reviewer’s concern, we have largely expanded the information in procedure, particularly in mindfulness-based family psychoeducation.
In specific, we have expended the second paragraph in procedure (line 243-249) and added a third paragraph (line 258-267) to explain the structure of the mindfulness-based family psychoeducation program. Besides, an appendix on the session themes for both mindfulness arm and psychoeducation arm at the end of the manuscript and interested researchers can have more information to replicate the study.
We expanded the data analysis section, stating that “We included participants who completed 50% of above to complete the post-test and follow-up for program evaluation” (in line 355). In this study, all 65 participants met this criteria as none of them have absent for more than two sessions.
In addition, it could also be interesting to look for significant differences between the participants in the different experimental groups with respect to the dependent variables considered.
Our responses: in this revised manuscript, we included subgroup analysis in section 3.4. We found gender and education of caregiver as predictors of outcome. We included a new paragraph to discuss the finding in subgroup analysis (line 527-535).
Without knowing what type of intervention was carried out, for how long, and without assessing further differences in the sample, I do not think it is possible for me to pass judgement on the work. With regard to the introduction, on the other hand, I think it is well written, although I would suggest dividing it into paragraphs and especially giving more prominence to the part concerning cultural differences in EE. Regarding discussion and conclusion, I believe that Authors could be find more reasons why the intervention showed limited effectiveness after considering the suggested indicators.
Our responses:
We have made the following revisions accordingly:
1) As mentioned, we have largely expanded the information that explained the intervention procedure.
2) we expanded the concern of EE in family intervention and pointed out that previous Chinese family intervention studies did not include EE as mediator or outcome measure. It adds significance to this study.
3) since our finding is consistent with a recent study in Japan, indicating a family psychoeducation program showed effective for caregivers of chronic patients but not for caregivers of patients with onset in last five years. We emphasized this evidence in discussion and conclusion and suggests further study should target caregivers of chronic patients (line 461-465, 478-480). We also recommend further study should increase the dosage, based on the instructor’s feedback, ensuring caregivers can benefit from a caregiver intervention.
Thank you very much for your helpful comments and they have largely improved the quality of the manuscript.
Round 2
Reviewer 1 Report
The authors implemented my suggestion, and I believe the paper benefitted from this revision.
No other comments from my side
Author Response
Thank you for your effort in reviewing this manuscript.
Reviewer 2 Report
Dear Authors,
thank you for your work. I believe that many improvements have been made that have increased the work's scientific value. Perhaps I would add other elements to the discussion concerning the effectiveness in general of mindfulness psycho-education interventions on outcomes similar to those you have considered. This would make it possible to corroborate the usefulness of the training, coupled, however, with the limitations you have found regarding the specific sample considered.
Author Response
Thank you for your comments. I have revised the first paragraph of the section of discussion, based on your thoughtful suggestions. I have further refined the language in the conclusion.
Besides, I have sent the manuscript to the professional editing service.